# Nanoscale protein architecture of the kidney glomerular basement membrane

Hani Suleiman[1], Lei Zhang[1], Robyn Roth[2], John E Heuser[2], Jeffrey H Miner[2,3]*, Andrey S Shaw[1,4]*, Adish Dani[1,5]

[1]Department of Pathology and Immunology, Washington University School of Medicine, St. Louis, United States; [2]Department of Cell Biology and Physiology, Washington University School of Medicine, St. Louis, United States; [3]Renal Division, Department of Medicine, Washington University School of Medicine, St. Louis, United States; [4]Howard Hughes Medical Institute, Washington University School of Medicine, St. Louis, United States; [5]Hope Center for Neurological Disorders, Washington University School of Medicine, St. Louis, United States

**Abstract** In multicellular organisms, proteins of the extracellular matrix (ECM) play structural and functional roles in essentially all organs, so understanding ECM protein organization in health and disease remains an important goal. Here, we used sub-diffraction resolution stochastic optical reconstruction microscopy (STORM) to resolve the in situ molecular organization of proteins within the kidney glomerular basement membrane (GBM), an essential mediator of glomerular ultrafiltration. Using multichannel STORM and STORM-electron microscopy correlation, we constructed a molecular reference frame that revealed a laminar organization of ECM proteins within the GBM. Separate analyses of domains near the N- and C-termini of agrin, laminin, and collagen IV in mouse and human GBM revealed a highly oriented macromolecular organization. Our analysis also revealed disruptions in this GBM architecture in a mouse model of Alport syndrome. These results provide the first nanoscopic glimpse into the organization of a complex ECM.

*For correspondence: shaw@pathology.wustl.edu (ASS); jminer@dom.wustl.edu (JHM)

Competing interests: The authors declare that no competing interests exist.

## Introduction

Kidney glomeruli are specialized capillary tufts responsible for filtration. They contain the glomerular filtration barrier (GFB), an intricate structure that constitutes the permselective barrier between the bloodstream and the urine. The GFB is organized into three major layers: (1) endothelial cells with fenestrations that line the glomerular capillaries; (2) unique epithelial cells called podocytes that extend many 'foot' processes that interdigitate with those of adjacent podocytes and are linked to each other via a specialized cell/cell junction called the slit diaphragm; and (3) the glomerular basement membrane (GBM), which is composed of extracellular matrix (ECM) proteins secreted by the endothelial cells and podocytes that flank it (*Miner, 2011*). The GFB's major physiological role is to allow the unimpeded passage of water and small solutes (including metabolic waste products) from the bloodstream into the urinary space, while restricting the passage of plasma proteins such as albumin and immunoglobulins. Although the exact mechanism whereby the GFB functions in this manner is the subject of much controversy, it is clear that injuries or genetic defects in any one of the three layers can cause GFB malfunction that results in increased levels of protein in the urine, a hallmark of diverse kidney diseases. In this study we focus on the ultrastructure and molecular organization of the GBM and its connections to podocytes and endothelial cells.

The GBM contains specific isoforms of laminin (laminin α5β2γ1, or LM-521), type IV collagen (primarily the collagen α3α4α5(IV) network), heparan sulfate proteoglycan (HSPG) (primarily agrin), and nidogen (*Miner, 2012*). Laminin and type IV collagen self-polymerize into networks that are connected to each

**eLife Digest** The blood that flows through the body must be continually filtered to remove waste products and to ensure that it contains optimal levels of water and salts. Filtration is performed inside the kidneys by tufts of small blood vessels called glomeruli. These glomerular capillaries allow water and waste products to pass from the blood into the urine, while holding back proteins and blood cells. The wall of a glomerular capillary consists of two layers of cells flanking a third layer called the glomerular basement membrane. If any of these layers malfunctions, it becomes possible for proteins to pass into the urine. This is a clear sign of kidney disease.

The basement membrane is composed of proteins secreted by the two layers of cells, but little was known about how these proteins are organized. Now, Suleiman et al. have adapted a new form of high-resolution optical microscopy called STORM to study the structure of the glomerular basement membrane in both mouse and human kidney tissue. By combining data from STORM and electron microscopy, Suleiman et al. showed that the proteins in the glomerular basement membranes of both species are arranged similarly to form a distinctive layered structure. This suggests that the organization of the basement membrane plays a critical role in its function.

The technique was used to demonstrate that proteins were not organized in the glomerular basement membrane in tissue samples taken from mice suffering from Alport syndrome, a genetic disorder of the kidneys. In addition to suggesting that the disorganization of basement membranes may play an important role in disease, this work also provides a method for investigating the structure of the basement membrane in diverse types of tissue.

other and/or to cell surface receptors by HSPGs and nidogen; in addition, some receptors can bind directly to laminin or type IV collagen (*Yurchenco et al., 2004*). A great number of biochemical studies of ECM proteins have revealed the molecular basis for their self-assembly and interactions with each other and with cellular receptors (reviewed in *Yurchenco, 2011*), and detailed structures are available for specific domains of several ECM proteins (*Khoshnoodi et al., 2006*; *Carafoli et al., 2012*). However, a systematic analysis of the actual spatial arrangement of ECM components within basement membranes, with respect to each other and to their receptors, has not yet been performed in vivo.

Studying the molecular architecture of ECM protein arrangement poses particular challenges in the context of the GBM. First, the GFB with the GBM cannot yet be reconstituted in vitro. Second, the ~200-nm thickness of a typical mouse GBM is close to the diffraction limited resolution of conventional light microscopy, thus preventing its use to analyze GBM ultrastructural and macromolecular organization. Transmission electron microscopy (EM) studies show electron dense and translucent layers (lamina densa and lamina lucida), but it is unclear whether these reflect an organization of proteins in the GBM (*Chan et al., 1993*). The ability to decipher ECM protein organization thus will require methods with nanometer resolution and high efficiency molecular labeling.

Recently, the invention of 'super-resolution' light microscopic approaches have broken the diffraction limit of light microscopy, combining a nanoscopic image resolution with several advantages of fluorescence microscopy, such as high efficiency and multi-color labeling (reviewed in *Huang et al., 2009*; *Ji et al., 2008*; *Hell, 2009*). In particular, single molecule localization-based imaging methods such as (fluorescence) photoactivable localization microscopy ((F)PALM) and stochastic optical reconstruction microscopy (STORM) (*Betzig et al., 2006*; *Hess et al., 2006*; *Rust et al., 2006*) achieve a molecular scale resolution that can bridge our understanding of supramolecular complexes from the molecular/biochemical to the cellular scales. The strength of various sub-diffraction microscopy methods to dissect the nanoscale architecture of multi-protein complexes has been exemplified recently in the case of neuronal synapses (*Dani et al., 2010*), adhesion complexes (*Kanchanawong et al., 2010*), centrosomes (*Lau et al., 2012*; *Lawo et al., 2012*; *Mennella et al., 2012*) and the nuclear pore complex (*Szymborska et al., 2013*).

Given the importance of ECM proteins in tissue morphogenesis and in diverse human diseases (*Miner and Yurchenco, 2004*), in this study we investigated the ECM protein organization within the GBM. Using a panel of antibodies, we performed 3D, multichannel STORM in kidney sections and determined the positions of well-defined protein epitopes with nanometer precision. Our data reveal that the GBM is composed of two layers of laminin and agrin molecules, with their N terminal domains

oriented towards the center of the GBM. Using transgenic mice expressing a human laminin chain we observed a striking difference in laminin dynamics within the GBM. Results from our imaging approach provide an in situ model of GBM supramolecular organization and align remarkably well with biochemical interactions gleaned from in vitro studies. Finally, we observed distinct distributions of collagen IV molecules and their disruption in a mouse model of Alport syndrome, a hereditary disease of the GBM caused by collagen IV mutations.

## Results

### Imaging the glomerular filtration barrier ultrastructure with STORM and STORM/deep-etch electron microscopy correlation

To reveal supramolecular organization within the GBM, we performed STORM on glomeruli in kidney sections using antibodies conjugated to Alexa 647, a bright fluorescent photoswitchable dye (*Bates et al., 2007*; *Dani et al., 2010*). The dense protein network within the GBM results in a higher fluorescence background and significant light scattering. This poses a major hurdle in obtaining high resolution STORM data from kidney sections. We therefore optimized the fixation and tissue sectioning methods extensively. These preliminary studies established that a modification of the Tokuyasu method of cryo-embedding and cryosectioning at 200 nanometers (nm) thickness was optimal (*Tokuyasu, 1973*).

We began with studies of agrin, a ~200 kDa protein that is the major HSPG component of the GBM (*Groffen et al., 1998*). Agrin's N-terminus (agrinN) binds to laminin via the laminin γ1 chain's coiled-coil domain (*Kammerer et al., 1999*), whereas agrin's C-terminus (agrinC) binds to integrins and dystroglycan (*Singhal and Martin, 2011*), receptors present on both podocytes and endothelial cells. To investigate agrin's organization in the GBM, we first labeled kidney sections with an antibody to agrinC. In contrast to conventional immunofluorescence images of agrin, STORM resolved two distinct layers of agrinC within the GBM (*Figure 1A,B*). To quantitatively document agrinC's distribution, we digitally selected multiple regions across glomeruli. AgrinC localizations in each region were fitted with a double Gaussian distribution, and the mid-position between the peaks was identified from each image and used as a reference to align subsequent regions. The cumulative localizations obtained from 80 regions are displayed as a histogram (*Figure 1C*). The gap between agrinC layers, measured as the peak-to-peak distance, showed a mean of 137.9 nm ± 1.9 nm standard error of mean (SEM) for the 80 regions; the distance was 132.7 nm (±0.9 nm) across more than 600 GBM regions (*Figure 1D*).

We next imaged the relationship between the GBM and the flanking podocytes and endothelial cells by double labeling with antibodies to agrinC and to the sialoprotein podocalyxin, which is expressed on the surfaces of both endothelial cells and podocytes (*Kerjaschki et al., 1984*). Dual channel STORM imaging showed that the two layers of agrinC were indeed localized between two layers of podocalyxin (*Figure 1E,F* and *Figure 1—figure supplement 1*). To further correlate the STORM molecular localizations, we developed a hybrid STORM-electron microscopy (EM) approach (see schematic in *Figure 1—figure supplement 2*). After performing STORM, tissue sections attached to the cover glass were subjected to quick-freezing followed by platinum deep-etching (*Heuser, 1981*). The tissue and cover glass underlying the platinum replicas were dissolved, and the replicas were transferred to a grid for examination by EM (*Figure 1G*). Superimposition of the podocalyxin-agrinC STORM image with deep-etch EM confirmed that podocalyxin labeled endothelial cells as well as podocytes, while agrin was localized to the GBM (*Figure 1H*). The STORM–EM correlation also confirmed that ultrastructural features of the GFB such as podocyte foot processes and the GBM were well-preserved.

Lastly, we confirmed these results using immunogold labeling and electron microscopy (EM). Kidney Tokuyasu sections labeled with anti-podocalyxin and anti-agrinC were initially detected using gold-conjugated secondary antibodies followed by deep-etch EM. Similar to its STORM localizations, anti-podocalyxin labeled both podocyte foot process and endothelial membranes (*Figure 1I*), but initial attempts to detect agrinC resulted in very sparse labeling. We postulated that the gold-labeled secondary antibody might not efficiently penetrate the GBM. To circumvent this problem, we used a third antibody (goat anti-rabbit) to bridge the primary antibody (rabbit anti-agrinC) and the gold-conjugated antibody (anti-goat). Images generated using this method confirmed the two agrin layers in the GBM (*Figure 1J*). These data validate the STORM localizations obtained on kidney sections and also confirm the challenges associated with achieving high efficiency immunogold labeling on a highly cross-linked structure such as the GBM.

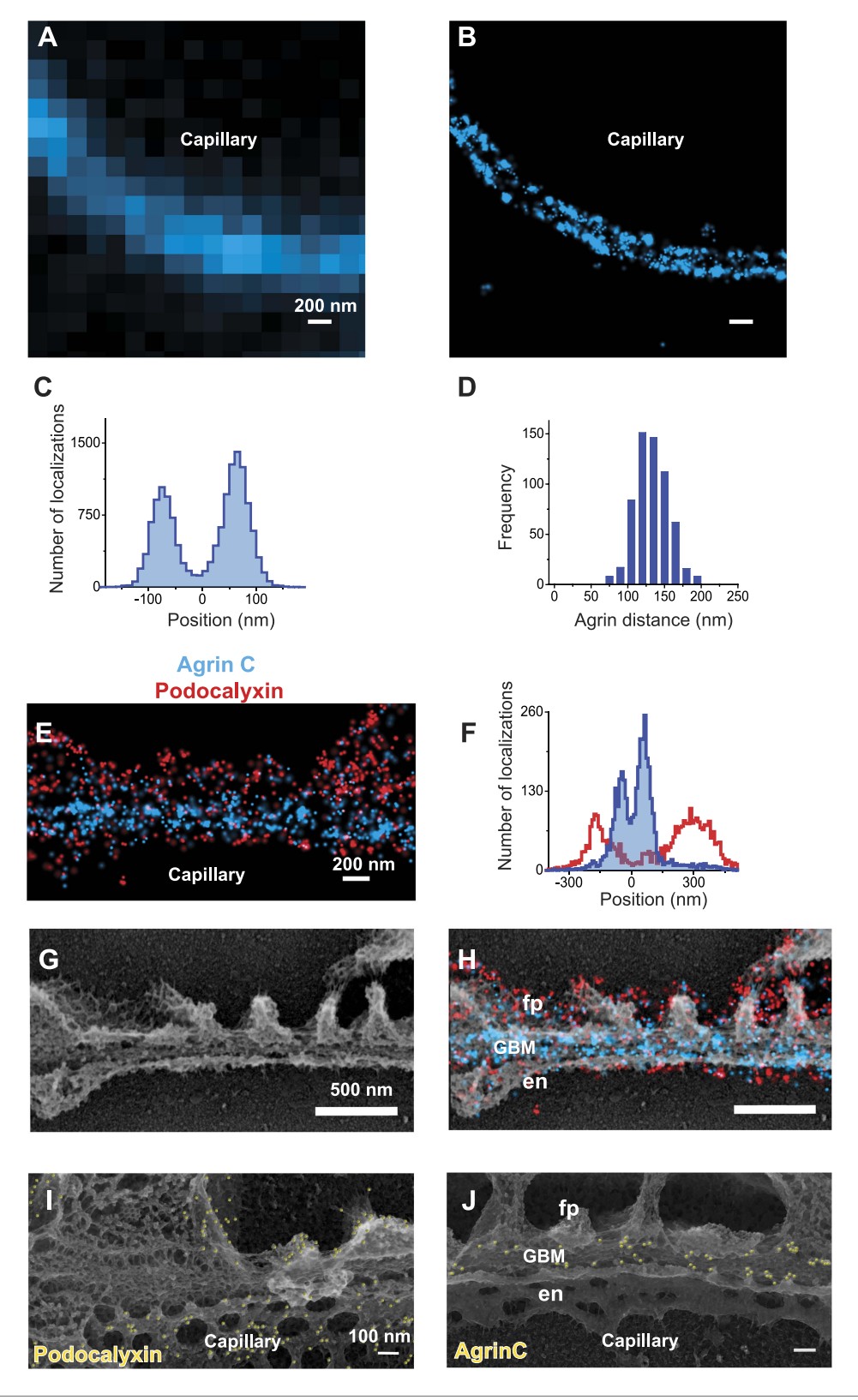

**Figure 1**. STORM and STORM-EM image correlation of the mouse GBM. Conventional fluorescence (**A**) and STORM (**B**) images of a kidney glomerular capillary loop labeled with an antibody to agrin. (**C**) Projection histogram of agrin STORM localizations accumulated from 80 different capillary regions. (**D**) AgrinC peak-to-peak distance

*Figure 1. Continued on next page*

*Figure 1. Continued*

measured across 604 regions in this study. (**E** and **F**) 2-color STORM and quantification of agrinC and podocaylxin labeling along a capillary region. (**G**) Platinum deep etch replica prepared from the section shown in (**E**) and imaged by EM. (**H**) Overlay of the STORM and EM images shows ultrastructural features such as podocyte foot processes (fp), endothelial cells (en) and the GBM. Podocalyxin labeling is seen along the foot process periphery and agrin localization within the GBM. *Figure 1—figure supplement 1* shows a wide field podocalyxin-agrin STORM image overlayed with its EM correlation and *Figure 1—figure supplement 2* shows a schematic overview of the STORM–EM correlation procedure. (**I** and **J**) Immuno-gold labeling and platinum replica EM from a kidney section confirm podocalyxin localization to the fp and en sides (**I**) and agrinC localization in two layers in the GBM (**J**). *Figure 1—figure supplement 3* shows a STORM image and histogram of two separate antibodies labeling the C-terminus end of agrin.

The following figure supplements are available for figure 1:

**Figure supplement 1**. Low magnification image of EM/STORM correlation.

**Figure supplement 2**. Schematic showing steps involved in processing of samples.

**Figure supplement 3**. Similar pattern of staining from two different agrin antibodies.

## Positioning molecules within the GBM

Having established methods for tissue preparation, STORM imaging, and EM correlation, we decided to analyze the molecular organization of the GBM by comparing the positions of various ECM components to each other. Taking advantage of the robust and bimodal distribution of agrinC, we used a localization scheme in which the center position between the two agrinC layers was set as the origin, and the position of a second protein, imaged along with agrinC by 2-color STORM, was plotted on a reference frame that was marked by the agrinC positions at −68.9 and +68.9 ± 0.9 nm towards the endothelial and podocyte sides, respectively. By iterating this procedure across multiple regions and glomeruli, we could determine the position of various protein epitopes across the GBM with a high degree of precision. The various ECM proteins and their epitopes mapped in this study are illustrated in *Figure 2—figure supplement 1*.

### Agrin

We tested the validity and the sensitivity of our approach by using two additional antibodies against agrin. First, we found that the localization and distribution of two different antibodies to agrinC were nearly identical (*Figure 1—figure supplement 3*). In contrast, co-labeling with antibodies to agrinN and agrinC showed that agrinN was present in two layers that were shifted slightly internally to the two agrinC layers, with center positions at −57.5 ± 2.8 nm and 50.2 ± 3.28 nm (*Figure 2A*). This suggests that agrin is oriented either perpendicular or oblique to the plane of the GBM with the C-terminus nearest the adjacent cells, and the N-terminus, which binds the coiled-coil domain of laminin γ1, closer to the center of the GBM. The orientation and mapping of agrinC and agrinN to different locations within the GBM is consistent with the distinct molecular interactions of the two agrin domains as well as the ~95 nm extended conformation of agrin visualized by EM (*Denzer et al., 1998*).

### Integrin β1

We next investigated the position of integrin β1, the major integrin β subunit expressed by both podocytes and endothelial cells (*Kreidberg and Symons, 2000*). Integrin β1 interacts with agrinC (*Martin and Sanes, 1997*) (and with laminin; see below). Using an antibody that recognizes an extracellular epitope (*Bazzoni et al., 1995*), we observed two layers of integrin β1 that showed considerable overlap with agrinC (*Figure 2B*). The center positions of each integrin layer were calculated at -79.6 ± 3.3 nm towards the endothelial side and 77.1 ± 3.4 nm towards the podocyte side. The histograms as well as the center positions of integrin β1 indicate a distribution that closely overlaps with agrinC, with a slight outward shift towards the cell membranes.

### Laminin-521

LM-521 is a cross-shaped heterotrimer that polymerizes in the ECM via interactions among the N-terminal (LN) domains of the α5, β2, and γ1 chains found at the ends of the short arms (*McKee et al., 2007*). LM-521 binds to cell surface receptors, such as integrin α3β1, via the laminin globular

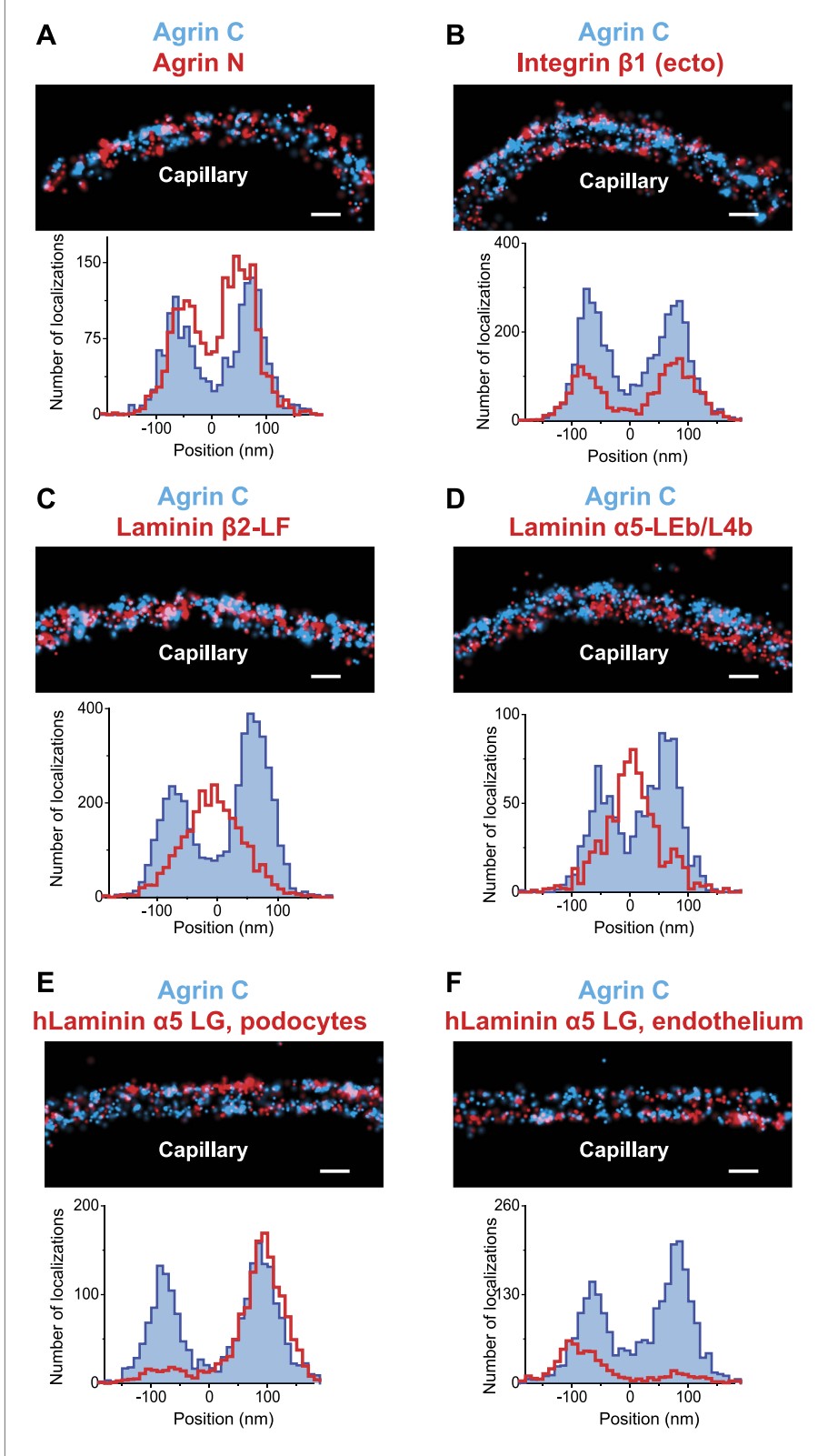

**Figure 2**. Positioning molecular domains within the GBM. Two channel STORM of agrinC along with antibodies to agrinN (**A**), the integrin β1 extracellular domain (**B**), laminin β2-LF domain (**C**) and laminin α5-LEb/L4b domain (**D**). Sections from transgenic mice expressing hLaminin α5 either from podocytes (**E**), or from endothelial cells (**F**)

*Figure 2. Continued on next page*

*Figure 2. Continued*

were labeled using an antibody against the human laminin α5-LG domain along with agrinC. Representative STORM images are displayed along with projection histograms of STORM localizations quantified across several regions. *Figure 2—figure supplement 1* illustrates domains of ECM molecules mapped in this study and *Figure 2—figure supplement 2* is a STORM image and histogram of laminin β2-LF along with integrin β1. Scale bar: 200 nm.

The following figure supplements are available for figure 2:

**Figure supplement 1**. Cartoon showing the relationship between antigenic epitopes and the structure of Agrin, Laminin and Collagen IV.

**Figure supplement 2**. STORM image of Integrin β1 and Laminin β2 confirms the central localization of Laminin.

(LG) domain of the α5 chain (*Ido et al., 2004*) (*Figure 2—figure supplement 1*). To localize and orient LM-521 in the GBM, we first used an antibody that recognizes the LF domain of laminin β2 (LMβ2-LF), which is near the center of the laminin heterotrimer. Co-labeling with anti-agrinC showed that LMβ2-LF localized between the agrinC layers at −3.2 ± 2.5 nm (*Figure 2C*). Co-labeling with anti-integrin β1 gave a LMβ2-LF position that was identical to that determined from the anti-agrinC co-labeling experiment (*Figure 2—figure supplement 2*) demonstrating the precision of the methodology. Next, we labeled the short arm of laminin α5 using an antibody raised against the LEb/L4b domain tandem (LMα5-LEb/L4b), also near the center of the heterotrimer. Similar to LMβ2-LF, the position of LMα5-LEb/L4b also mapped in the middle of the GBM, at −1.8 ± 3.6 nm (*Figure 2D*).

The span between the N- and C-termini of laminin α5 forms the long 'vertical' axis of the LM-521 cruciform trimer. Based on data from electron micrographs and crystal structures of various laminin domains, this axis is estimated to be 130 to 150 nm, while the horizontal axis, composed of the β2 and γ1 short arms, spans 70 to 80 nm (*Beck et al., 1990*; *Yurchenco and Cheng, 1993*; *Sasaki et al., 2004*). Our findings that the short arm epitopes from laminin α5 and β2 were located in the middle of the GBM prompted us to investigate the position of the C-terminal LG domain, which binds cell surface receptors such as integrins, dystroglycan, and the Lutheran cell adhesion molecule (*Miner, 2005*). Because we did not possess an antibody to the mouse LG domain, we made use of transgenic mice with doxycycline-inducible expression of human laminin α5, which assembles with mouse laminin β2 and γ1 to generate a functional chimeric human/mouse LM-521 trimer (*Goldberg et al., 2010*).

Transgenic mice in which human laminin α5 was expressed either from podocytes or from endothelial cells were analyzed. As endogenous laminin α5 has been shown to be synthesized by both cell types (*St John and Abrahamson, 2001*), this is a physiologically relevant approach. An antibody to the C-terminal LG domain of human laminin α5 (*Engvall et al., 1986*; *Tiger et al., 1997*) labeled the aspect of the GBM nearest the cell from which laminin α5 was secreted (*Figure 2E,F*). The positions of the LMα5-LG domain were calculated at 88.3 ± 3.07 nm towards the podocyte side and −84.7 ± 6.7 nm towards the endothelial side. Together with the data for LMα5-LEb/L4b (*Figure 2D*), this shows that LM-521 trimers are oriented with their C-terminal LG domains near the outer aspects of the GBM and their N-terminal domains towards the center of the GBM. The most straightforward interpretation is that the GBM's laminin component is divided into two networks, one produced by podocytes and the other by endothelial cells. The single band seen with the N-terminal antibody in the center of the GBM occurred because the two networks either overlap or are so close to each other that they are indistinguishable by STORM. In the vast majority of glomeruli, we observed that neither podocyte- nor endothelium-derived LM-521 crossed/flipped over, demonstrating that there is limited mobility of laminin trimers across the thickness of the GBM. A similar conclusion was also reported for the endogenous laminin trimer using a different method (*Abrahamson et al., 2007*).

## Collagen IV

Collagen IV chains contain a long collagenous domain composed of interrupted Gly-X-Y amino acid triplet repeats situated between non-collagenous N-terminal (7S) and C-terminal (NC1) domains (*Hudson et al., 2003*). These chains assemble intracellularly into ~400 nm long triple helical heterotrimers (also known as protomers) that are secreted into the ECM, where they polymerize via hexameric NC1 and dodecameric 7S interactions to form a chicken wire-like network one to several chains thick (*Khoshnoodi et al., 2008*). The main collagen IV network of the adult GBM consists of collagen

α3α4α5(IV) protomers, which are secreted solely by podocytes (*Abrahamson et al., 2009*). The details regarding how collagen IV protomers align in the healthy GBM are unexplored.

We used two antibodies to collagen α3α4α5(IV), one against the assembled hexameric C-terminal NC1 domains (*Heidet et al., 2003*) and a second against an epitope close to the N-terminus ('peri-N') of the collagen α5(IV) chain (*Kagawa et al., 1997*). STORM localization of these two disparate epitopes in comparison to agrinC showed that both collagen IV epitopes were broadly distributed within the GBM, with positions concentrated close to the center of the GBM, at 12.5 ± 7.2 nm for NC1 and 11.92 ± 3.4 nm for the peri-N epitope (*Figure 3A,B*).

In developing glomeruli, collagen α1α1α2(IV) is the major collagen IV isoform. In the mature GBM, α1α1α2(IV) is down regulated and mostly replaced by α3α4α5(IV) (*Miner and Sanes, 1994*). However,

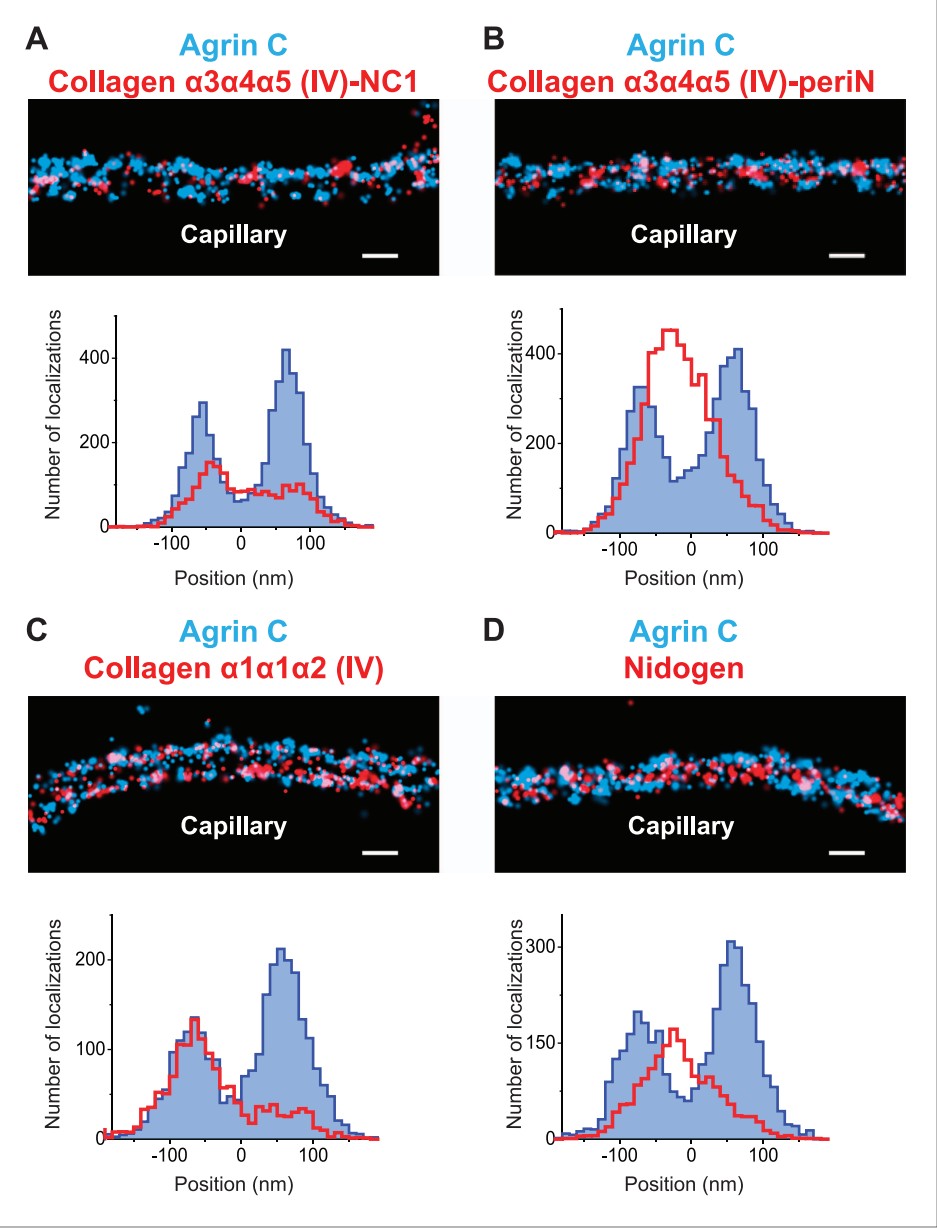

**Figure 3**. Collagen IV distribution in the GBM. STORM localization of collagen α3α4α5 (IV)-NC1 (**A**) and collagen α3α4α5 (IV)-periN (**B**) domains along with agrinC show the position of the collagen α3α4α5 (IV) network towards the center of the GBM. In contrast, collagen α1α1α2 (IV) labeling (**C**) is restricted to the endothelial side of the GBM. (**D**) Nidogen localizes in the center of the GBM. Scale bar: 200 nm.

since α1α1α2(IV) persists at a low level in the adult GBM and is prominent in most other endothelial BMs, we were interested to localize it in relationship to α3α4α5(IV). Collagen α1α1α2(IV) localized close to the endothelial side of the GBM with a center position at −59.5 ± 5.2 nm (*Figure 3C*), consistent with its known synthesis by endothelial cells (*Heidet et al., 2000*). Importantly, collagen α1α1α2(IV) and α3α4α5(IV) showed distinct distributions within the GBM.

## Nidogen

In all basement membranes, the collagen IV network is thought to be cross-linked to the laminin network by nidogen, a 150 kDa dumbbell-shaped glycoprotein that binds to the LEb domain of laminin γ1 and to the collagenous domain of collagen α1α1α2(IV) (*Aumailley et al., 1989*). This view of nidogen as a cross-linker, however, has recently been challenged (*Behrens et al., 2012*). Staining with anti-nidogen localized it to the center of the GBM, with a position of −0.22 ± 3.9 nm, close to both collagen α3α4α5(IV) and LM-521 epitopes (*Figure 3D*). While this is consistent with its role as a potential cross-linker, the lack of molecular details about the interactions between nidogen and collagen α3α4α5(IV) and a lack of information of the exact nidogen epitope studied limit any further interpretation. The positions of various mouse ECM protein epitopes are represented graphically in *Figure 4*.

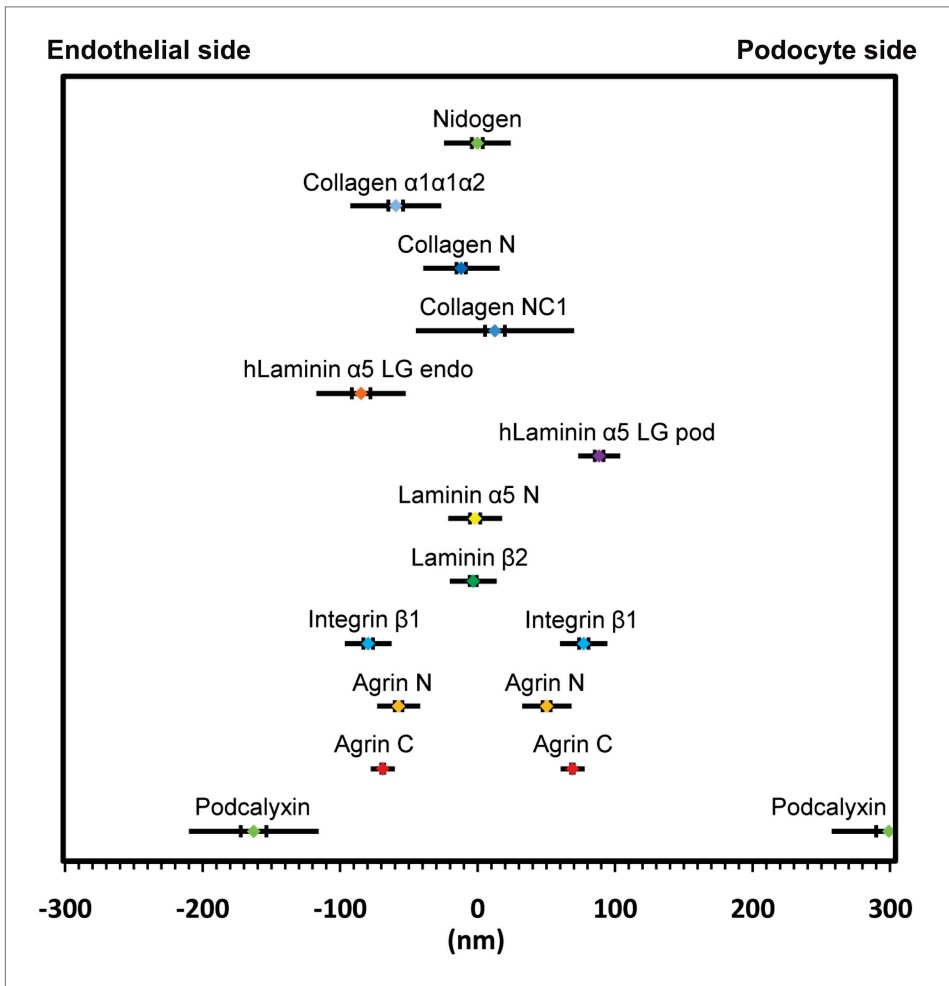

**Figure 4**. Positions of molecular epitopes within the GBM. A map of axial positions of various GBM protein domains obtained from STORM imaging in this study. For each protein, the colored dot specifies the mean axial position; the two vertical lines represent the standard error of the mean, and the half-length of the horizontal bar denotes the standard deviation.

The following source data are available for figure 4:

**Source data 1**. Calculated relative position of molecules in the mouse GBM.

## The organization of the human GBM

The human GBM seems to be composed of components that are orthologous to those in mouse, but it is at least two times thicker than the mouse GBM. We therefore attempted to map ECM protein epitopes within the human GBM to better understand its architecture and compare with the mouse GBM. We first labeled the integrin β1 ectodomain along with agrinC and observed that, similar to mouse GBM, both protein epitopes localized along the endothelial and podocyte surfaces of the human GBM (*Figure 5A,B*). It is notable that significantly more agrinC was detected at the podocyte vs the endothelial aspect (*Figure 5B*); this could be due to either epitope masking or differences in expression. The peak-to-peak distance of integrin β1 was measured to be 428 ± 14.4 nm, consistent with the known thickness of the human GBM. Given these data, and since two separate anti-integrin β1 antibodies showed similar distributions (*Figure 5—figure supplement 1*), we selected integrin β1 as a reference and mapped laminin and collagen IV epitopes. We observed that the anti-LMα5-LG epitope labeled two layers close to the endothelial and podocyte sides of the GBM in a location near the integrin β1 epitope (*Figure 5C*). However, unlike the mouse, where LMα5-LG was restricted to the two sides of the GBM, in human we also observed LMα5-LG towards the central aspect of the GBM. STORM–EM correlation confirmed that the human GBM was intact and the GBM central LMα5-LG localizations could not arise due a poorly preserved GBM or sectioning artifact (*Figure 5—figure supplement 2*). Given the inability of cell surface receptors to interact with LMα5-LG domains in these internal sites, it is not clear how internal LM-521 molecules would be organized. We speculate that LN domain interactions with the subepithelial and subendothelial LM-521 networks could be involved.

We next imaged collagen α3α4α5(IV) NC1 and peri-N epitopes along with integrin β1 in the human GBM (*Figure 5E,F*). Both collagen α3α4α5(IV) epitopes were localized at the center of the GBM, with a clear separation from integrin β1. Staining for collagen α1α1α2(IV) using an antibody to the N-terminal part of the collagenous domain (*Foellmer et al., 1983*) showed that it was localized to the endothelial side of the GBM (*Figure 5D*), consistent with previous results using conventional immunofluorescence microscopy (*Butkowski et al., 1989*). Further, co-labeling for α3α4α5(IV)-NC1 and α1α1α2(IV) epitopes showed that the two collagen networks had distinct adjacent distributions (*Figure 5—figure supplement 3*), with the α1α1α2(IV) network occupying the space between the central α3α4α5(IV) layer and the endothelial surface of the GBM. These data indicate that the increased thickness of the human GBM can be attributed to an expanded collagen α3α4α5(IV) network that is significantly wider than that in the mouse GBM together with an additional layer of LM-521.

## Distribution of GBM components in a mouse model of Alport syndrome

The distinct distributions of the components of the GBM prompted us to assess whether the organization of the GBM is disrupted in a specific disorder of the basement membrane. We therefore analyzed a mouse model of autosomal recessive Alport syndrome that lacks the collagen α3α4α5(IV) network due to a *Col4a3* null mutation (*Miner and Sanes, 1996*). In both humans and mice, the lack of this network results in a compensatory increase in the expression of collagen α1α1α2(IV), which is normally found at low levels in adult GBM (*Kashtan and Kim, 1992*; *Miner and Sanes, 1996*). Despite this compensation, the GBM becomes segmentally split and thickened, and this is associated with hematuria, proteinuria, and progressive renal failure (*Hudson et al., 2003*). To investigate the architecture of the GBM in this disease model, we examined the organization of agrinC and collagen α1α1α2(IV). In contrast to the wild-type GBM, the Alport GBM showed segments of capillary loops where the two layered organization of agrinC was disrupted, while in some segments it was intact (*Figure 6A,B*). The localization of collagen α1α1α2(IV) was also altered, with its distribution spread across the width of the GBM and no longer restricted to the endothelial side. This altered collagen α1α1α2(IV) distribution was evident even in capillary loop segments that showed a relatively preserved agrinC organization (*Figure 6C,D*). Thus, we conclude that an intact collagen α3α4α5(IV) network helps to maintain agrin and collagen α1α1α2(IV) organization in the healthy GBM.

## Discussion

Determining the structure, molecular interactions and spatial organization of ECM proteins are important steps towards understanding their roles in tissue function, morphogenesis and disease. Our study describes a novel super resolution fluorescence microscopy approach to reconstruct the molecular architecture of ECM proteins within the GBM. We developed a method to achieve systematic nanoscale molecule mapping in dense tissue sections using STORM and ultrastructural imaging on the same

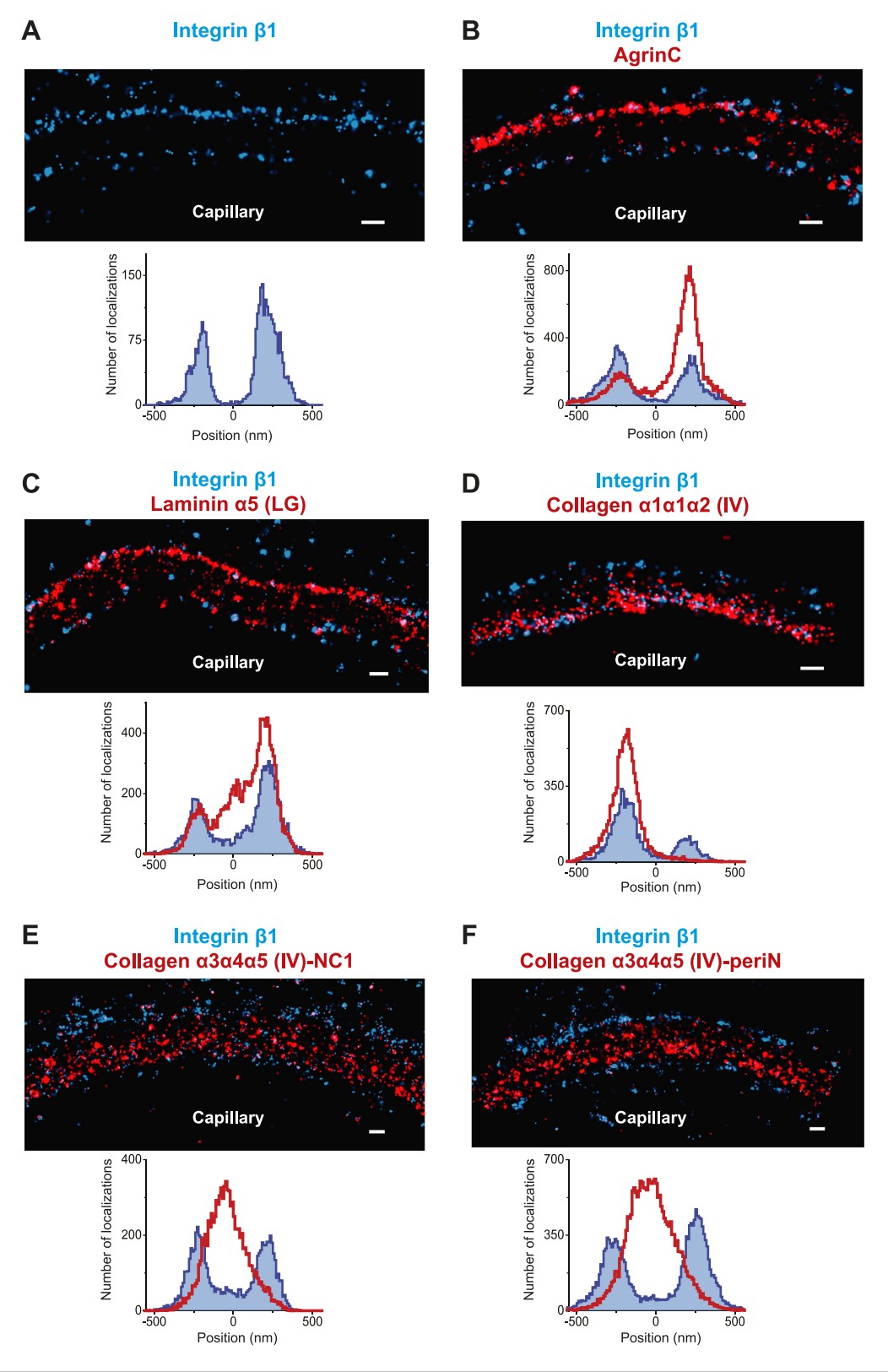

**Figure 5**. Molecular organization of the human GBM. (**A**) Single channel STORM of integrin β1-labeled human GBM sections. *Figure 5—figure supplement 1* shows two channel STORM using two different integrin β1 *Figure 5. Continued on next page*

*Figure 5. Continued*

antibodies. Human kidney sections were labeled with antibodies to integrin β1 ectodomain and: agrinC (**B**), laminin α5 (LG) (**C**), collagen α1α1α2 (IV) (**D**), collagen α3α4α5 (IV)-NC1 (**E**) and collagen α3α4α5 (IV)-periN (**F**). *Figure 5—figure supplement 2* shows anti-laminin α5-LG labeling of the human GBM and a replica EM image of the same section. *Figure 5—figure supplement 3* shows anti-collagen α1α1α2(IV) and collagen α3α4α5 (IV)-NC1 labeling of a human GBM region. Scale bar: 200 nm.

The following source data and figure supplements are available for figure 5:

**Source data 1**. Calculated relative position of molecules in the human GBM.

**Figure supplement 1**. Similar pattern of staining obtained with two different human integrin β1 antibodies.

**Figure supplement 2**. Laminin a5 stains multiple layers in normal human GBM.

**Figure supplement 3**. Distinct localization of Collagen α1α1α2 (IV) and Collagen α3α4α5 (IV) in human GBM.

samples using EM. The robust and relatively rapid throughput of this methodology allowed us to reconstruct the molecular architecture of the mouse as well as human GBMs.

## Nanoscopic imaging in tissues

Light microscopic imaging shows the components of the GBM as diffusely distributed, suggesting either an amorphous structure or an organization below the resolution of conventional light microscopy. While transmission EM imaging of the glomerulus with $OsO_4$ demonstrates an electron dense layer in the GBM, the significance of this finding has been unclear. Immunogold-EM has been deployed to study ECM protein organization (*Miosge et al., 1999*), but constraints of sample preparation, antibody accessibility, and quantitation prevent its extensive application. In our study, we performed STORM as well as EM and immunogold-EM using the same tissue and antibody labeling procedures. This allowed us to validate data using both methods and also illustrated the challenges associated with immunogold EM.

Super resolution fluorescence microscopic methods have now been widely used to resolve subcellular structures in cultured cells, but the use of these methods to analyze tissue has been challenging. Single molecule probe based methods (F)PALM and STORM were originally developed using photoswitchable fluorescent proteins (*Betzig et al., 2006*; *Hess et al., 2006*) or photoswitching properties of fluorescent dyes (*Rust et al., 2006*). Exogenous expression of multiple photoswitchable fusion proteins and their functional validation is not yet feasible for studying multi-protein complexes in mammalian tissues. Using a number of well characterized antibodies provides a practical alternative for tissue imaging. Thus, we focused on using photoswitchable dye-conjugated antibody labeling for our experiments. Fluorescence imaging of tissue also presents special challenges, mainly regarding the methods of fixation, labeling, sectioning and the method of electron microscopic correlation. Sub-diffraction microscopy methods typically map the positions of labeled proteins; however, putting these in the context of membranes and other ultrastructures requires EM. Recently, a few studies have described super-resolution imaging of photoswitchable fusion proteins expressed in cultured cells and in *C. elegans*, followed by plastic embedding and EM correlation (*Betzig et al., 2006*; *Watanabe et al., 2011*; *Kopek et al., 2012*). Conventional fluorescence microscopy of resin embedded tissue sections labeled with antibodies has also been described (*Micheva and Smith, 2007*). Here, we tested a variety of fixation methods, sectioning thicknesses and electron microscopic approaches. Our best results were obtained when we collected 200-nm thickness Tokuyasu cryosections directly onto a coverglass. Since the Tokuyasu method was originally developed for the preparation of tissue for EM imaging, we were able to use freeze etch EM after STORM, to produce correlated images. We believe that the method described here is a breakthrough as it should be useful for the imaging of a wide variety of other tissues.

## Molecular architecture of the GBM

The results from our study demonstrated that the ECM within the GBM, far from being amorphous, is a highly structured and laminated amalgam of interacting protein networks (*Figure 4*). We showed that agrin and laminin-521 each form two layers within the mouse GBM. In each layer, both proteins

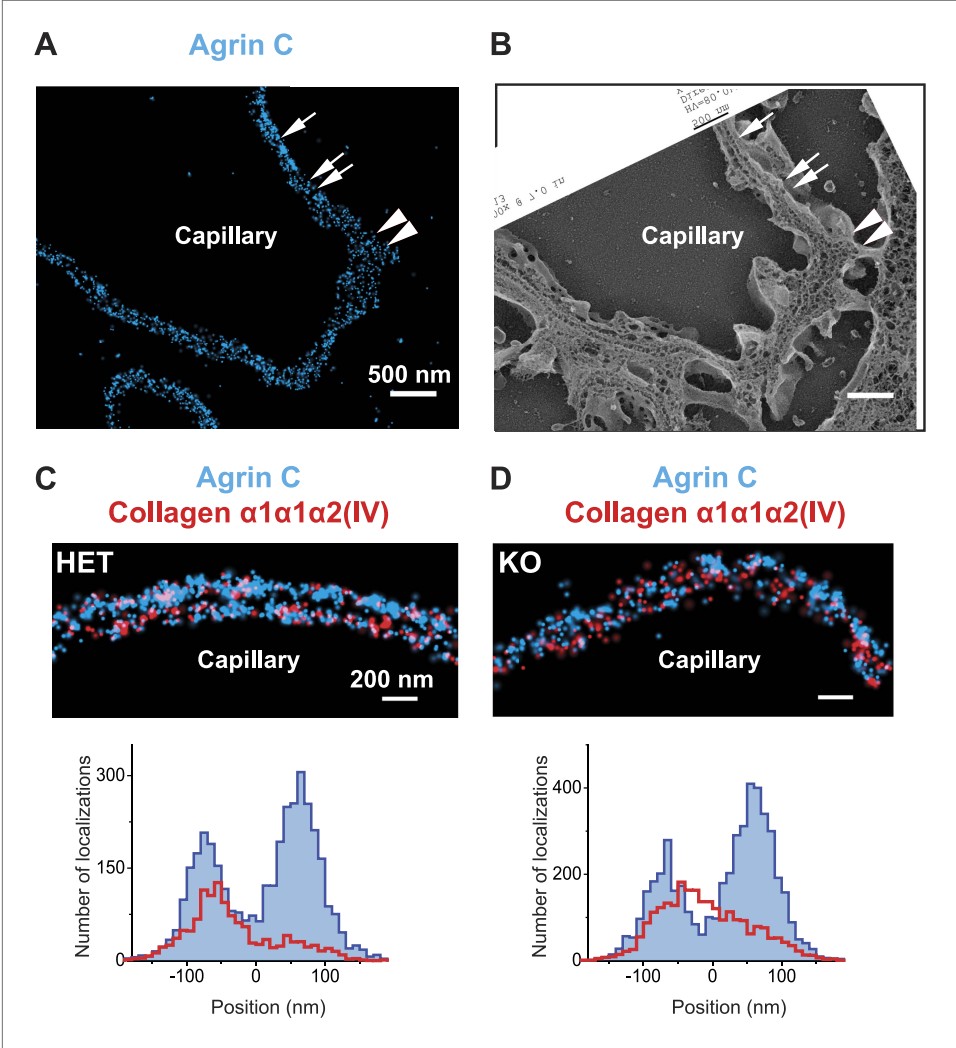

**Figure 6**. Breakdown of the GBM molecular architecture in a mouse model of Alport syndrome. (**A** and **B**) STORM and EM images of a capillary loop from a collagen α3(IV) knockout (KO) mouse kidney labeled with agrinC shows a thin GBM with 2-layered agrin (single arrow) as well as a breakdown of the 2-layered agrin labeling pattern (double arrow) and a thick irregular GBM stretch showing disorganized, diffuse agrinC labeling (arrowheads). (**C** and **D**) Images and quantification of capillary loops selected from collagen α3(IV) KO and heterozygous (HET) kidney that show two layers of agrinC. Despite the intact agrin layers, collagen α1α1α2(IV) shows an atypical distribution spread across the GBM in the KO vs a single peak in the HET littermate control.

were oriented with their C-terminal LG domains close to the adjacent cell membranes, whereas the more N-terminal laminin short arm epitopes and the agrin N-terminus were localized towards the interior of the GBM. Cell culture and biochemical data support these observations: the LG domains of laminin α5 and agrin are known to interact with cell surface receptors (*Martin and Sanes, 1997*; *Yurchenco et al., 2004*; *Kikkawa et al., 2007*), and the agrin N-terminal domain interacts with laminin γ1 near the center of the coiled-coil domain (*Kammerer et al., 1999*). But the ~140 nm vertical axis of laminin must be reconciled with our definitive demonstration in mouse that there are two separate networks that meet in the middle of the GBM; the ~200 nm wide mouse GBM cannot contain two fully extended laminin trimers, which would approach a 300 nm span. Models predicting either a bent laminin conformation (*Yurchenco et al., 2004*) or an extended laminin situated at an oblique angle relative to the adjacent cells would be consistent with our data.

One of our most important and unexpected findings was the mapping of the collagen α3α4α5(IV) N- and C-terminal domains to near the center of the GBM, a position that in human GBM is too distant

from the positions of the integrin β1 extracellular domain (extending from both podocytes and endothelial cells) for a meaningful ligand–receptor interaction. In contrast, both agrinC and LMα5LG localized near integrin β1, implicating them as the likely biologically relevant ligands. At the endothelial aspect of the GBM, collagen α1α1α2(IV) was near enough to the cell to be capable of binding integrin β1. In our Alport syndrome mouse model, which lacks the collagen α3α4α5(IV) network, collagen α1α1α2(IV) was found at increased levels throughout the width of the GBM. One implication of this is that podocytes should be exposed to unfamiliar collagen IV ligands, which might then lead to podocyte injury and the glomerulosclerosis that is observed in the disease.

The GBM's collagen α3α4α5(IV) is secreted solely by podocytes (*Abrahamson et al., 2009*), and our results show that it eventually becomes concentrated near the center of the GBM, away from the podocytes. The secreted collagen α3α4α5(IV) protomers are thus likely able to permeate through the LM-521 and agrin networks adjacent to the podocyte, against the flow of filtrate, to reach the center of the GBM. Given the greater thickness of the human vs the mouse GBM, this process should either be faster or persist for a longer period in human glomeruli. In contrast, the collagen α1α1α2(IV) protomers made by endothelial cells form a network closely juxtaposed to the endothelium. These different collagen IV protomer behaviors may be fundamental to establishing and maintaining the GBM and could be related to its splitting and thickening in the setting of Alport syndrome.

## Materials and methods

### Tissue handling and genetically altered mice

Kidneys were isolated from mice after transcardial perfusion with phosphate buffered saline (PBS) containing 4% (wt/vol) Paraformaldeyde (PFA; EM Sciences, Hatfield, PA). After dissection, kidneys were cut into smaller pieces and fixed overnight with 4% PFA, followed by washing off excess PFA with PBS. Kidney pieces were immersed overnight at 4°C in a cryoprotectant solution of 2.3 M sucrose + 10% polyvinylpyrrolidone (PVP) in 0.1 M PIPES (pH = 7.2). Cryoprotected tissues were mounted on a metal sectioning pin and frozen by immersion in liquid nitrogen.

All transgenic and knockout mice used have been previously described. These included tetO$_7$-regulated human Laminin α5 cDNA (*Goldberg et al., 2010*), Nphs2-rtTA (*Shigehara et al., 2003*), Tie2-Cre (*Koni et al., 2001*), *Rosa26*-LoxP-neo-LoxP-rtTA (*Belteki et al., 2005*), and *Col4a3* null (*Miner and Sanes, 1996*) mice. Induction of human laminin α5 expression was achieved by feeding pregnant females doxycycline chow (0.15%) beginning when pregnancy was apparent and continuing after birth and after weaning.

De-identified human kidney samples from individuals with no known history of kidney disease were obtained through the Washington University George O'Brien Center for Kidney Disease Research. Samples were fixed overnight in 4% paraformaldehyde and cryoprotected as described above.

### Sectioning and immunohistochemistry

To capture tissue sections, acid cleaned and air-dried No. 1.0 coverglass were carbon coated for 1 min at a pressure of $2 \times 10^{-6}$ mbar. Carbon coated coverglass were glow discharged at $2 \times 10^{-2}$ mbar immediately before collecting tissue sections. Frozen tissues were sectioned at ~200-nm thickness on a Leica EM-FC6 ultracryomicrotome equipped with a diamond knife and sections were collected on the carbon coated coverslips. Sections were re-fixed for 20 min at room temperature (RT) with 4% PFA, followed by three washes in PBS and excess PFA was quenched using 50 mM glycine in PBS. Sections were further processed for immunolabeling in the following manner: (1) blocked overnight at 4°C using 2% bovine serum albumin (BSA) in PBS, (2) primary antibodies diluted in 2% BSA-PBS were applied overnight at 4°C followed by 3 × 20 min PBS washes at RT, (3) secondary antibodies diluted 3% BSA-PBS were applied at RT for 2–3 hr followed by PBS washes. (4) Immunolabeled sections were post-fixed using 3% PFA+ 0.05% Glutaraldehyde (EM Sciences) in PBS, washed in PBS in used for STORM.

### Antibodies

The primary antibodies used in this study and their approximate concentrations/dilutions used are shown in *Supplementary file 1*. Secondary antibodies for STORM were purchased from Jackson Immunoresearch and were custom conjugated to Alexa647 reporter dye and either Alexa405, Cy2 or Cy3 activator dyes as described before (*Bates et al., 2007*).

## STORM microscope setup and image acquisition

STORM setup and image acquisition scheme were similar to that described before (*Dani et al., 2010*), with the following modifications. Briefly, the STORM rig was constructed around a Nikon Eclipse TiE inverted microscope fitted with the Nikon perfect focus system for focus stabilization, a motorized stage (Marzhauser) and a 100X 1.4NA objective (Olympus UPLSAPO). Illumination lasers, 642 nm (Vortran), 561 nm, 488 nm (Coherent, Sapphire) and 405 nm (Coherent, Cube) were shuttered using an acousto-optical tunable filter (AOTF, Crystal technologies). Laser beams were combined, expanded, collimated and focused at the back focal plane of the 100X objective. Total internal reflection fluorescence (TIRF) illumination was achieved with an objective type TIRF geometry using a custom-built translation stage.

Sections immunolabeled on carbon coverglass were inverted onto a slide containing a drop of imaging buffer containing mercaptoethylamine along with an oxygen scavenger system, and coverglass edges were sealed with nail polish. In all experiments described here, we performed single reporter (Alexa647)-multiple activator STORM as described before (*Bates et al., 2007*; *Dani et al., 2010*). Alexa647 images (~10,000 images per channel) acquired by the same objective were separated using a quad band dichroic ZT405/488/561/640rpc and filtered using ET705/72m emission filter (Chroma). Images were captured on an EM-CCD camera (iXon+ DU897, Andor) and analyzed using custom software. Image stacks were fitted with an elliptical Gaussian function to determine the centroid positions of fluorescent pixel intensity peaks. These positions termed 'STORM localizations' were rendered as STORM images or analyzed further quantitatively.

## Quantification of STORM localizations and molecule positions

To quantify STORM localizations and to estimate molecule positions with high precision, multiple GBM regions, each as a ~800 nm wide window, were selected towards the most peripheral aspect of circular capillary loops, to avoid the mesangial regions where the mesangial matrix meets the GBM. Selected regions were rotated to uniformly orient the podocyte and endothelial sides. STORM localizations from agrinC and a co-labeled molecule, were projected onto the perpendicular axis for each region and fitted with Gaussian functions to identify the centroid positions of the STORM localizations. The midpoint between the two agrinC centroid positions was set as zero and the position of the second molecule of interest was identified by the distance of its centroid position with respect to the zero. This procedure was iterated over multiple regions to estimate the mean position, standard error of the mean and standard deviation for each molecule. To view the quantitative profile of STORM localizations, a histogram was generated from each region by projecting localization points onto a line perpendicular to the long axis of the capillary loop. Each histogram was shifted to align them by their zero position and an accumulated histogram from several regions was constructed by adding up all localizations. The number of regions used to generate the histograms, the mean position of each molecule, standard error of the mean and standard deviation are reported as *Figure 4—source data 1* and *Figure 5—source data 1*. All quantifications were performed using custom scripts in Matlab and the data were plotted in Origin software.

## Quick-freeze, deep-etch electron microscopy and STORM-EM correlation

Quick-freeze deep-etch EM was performed according to published protocol, with minor modifications (*Heuser, 1980*). After STORM, nail polish from the coverglass was carefully removed by immersing in PBS, and the tissue sections were fixed in 2% glutaraldehyde in100 mM NaCl, 30 mM HEPES and 2 mM CaCl, pH 7.2 (NaHCaCl) at room temperature. 3 × 3 mm areas of the coverglass containing STORM imaged sections were cut, rinsed in dH$_2$O and frozen by abrupt application of the sample against a liquid helium cooled copper block with a Cryopress freezing machine. Frozen samples were transferred to a liquid nitrogen cooled Balzers 400 vacuum evaporator, etched for 20 min at −80°C and rotary replicated with ~ 2 nm platinum deposited from a 20° angle above the horizontal, followed by an immediate ~10 nm stabilization film of pure carbon deposited from an 85° angle. Replicas were floated onto a dish of concentrated hydrofluoric acid and transferred through several rinses of dH20 with a loopful of Photo-flo, picked up on Luxel grids (Luxel, Friday Harbor, WA), and photographed on a JEOL 1400 microscope with attached AMT digital camera. The glomeruli imaged by EM were matched with the corresponding STORM images, and the two images were superimposed using Adobe Photoshop. The STORM–EM correlation procedure is illustrated in *Figure 1—figure supplement 2*.

Immunogold labeling was done on cryosections collected on a 3 × 3 mm coverglass followed by detection with 12 nm colloidal gold affinity purified secondary antibodies (Jackson Immuno Research) diluted 1: 15 in PBS/2% BSA over 2 hr at RT, rinsed with three 10 min washes of PBS, followed by fixation in 2% glutaraldehyde in NaHCaCl. Prior to freezing, coverglass was rinsed in $dH_2O$, frozen and platinum replicas made as described above.

## Acknowledgements

We thank Dr Wandy Beatty for help with sectioning and Drs Bogdan Borza, Takako Sasaki, Jo Berden, and Johan van der Vlag for sharing antibodies.

## Additional information

### Funding

| Funder | Grant reference number | Author |
| --- | --- | --- |
| National Instutute of Diabetes and Digestive and Kidney Diseases | R21DK095419, R01DK078314, RO1DK058366 | Jeffrey H Miner, Andrey S Shaw |
| Alport Syndrome Foundation | | Jeffrey H Miner |
| Howard Hughes Medical Institute | | Andrey S Shaw |
| National Institute of Mental Health | R21MH099798 | Adish Dani |

The funders had no role in study design, data collection and interpretation, or the decision to submit the work for publication.

### Author contributions

HS, AD, Conception and design, Acquisition of data, Analysis and interpretation of data, Drafting or revising the article; LZ, Quantified STORM localizations; RR, Performed deep etch EM; JEH, Conception and design, Acquisition of data, Analysis and interpretation of data; JHM, ASS, Conception and design, Analysis and interpretation of data, Drafting or revising the article

### Ethics

Animal experimentation: This study was performed in accordance with the recommendations in the "Guide for the Care and Use of Laboratory Animals" of the National Institutes of Health. Animals were handled according to approved institutional animal care and use committee (IACUC) protocol numbers 20100105, 20130059, and 20110282.

## Additional files

### Supplementary files

• Supplementary file 1. List of antibodies used in this study.

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
