## [Decision Letter]

Thank you for sending your work entitled “Nanoscale Protein Architecture of the Kidney Glomerular Basement Membrane” for consideration at *eLife*. Your article has been favorably evaluated by a Senior editor and 3 reviewers. The Reviewing editor has assembled the following comments to help you prepare a revised submission.

This manuscript provides novel insights into the organization of ECM in kidney glomerular basement membrane using correlative EM/super-resolution imaging. Using probes targeting different proteins (and in some cases different domains of the same protein) the authors investigate their relative distributions and show that the ECM in these cells is much more organized than generally believed. Overall this is a very well executed study with data of very high quality. In particular, it represents one of the few sub-diffraction imaging papers published to date, which utilized robust morphometric measurements. It is of interest to scientists involved in high resolution imaging, investigating the basement membrane, and nephrologists. However there are some issues that should be addressed before publication.

1) Explain more clearly why 3D-SIM and/or STED are not be suitable for imaging 200 nm sections. The authors should also consider validating some of their key results using another imaging method, like 3D-SIM and STED (and GFP fusion proteins) as it is likely to generate similar data.

2) Clarify the geometry of analysis in the text. While considerable detail is given in the Methods, the reader needs to understand the geometry of analysis as they read the results. Statements such as ‘selected multiple regions across glomeruli’ are uninformative and the definition of ‘localizations’ is lacking.

The reader should be informed of:

A) How different regions were chosen (that include both capillary and podocyte epithelia in approximate cross section).

B) The exact geometry of analysis (were lines joining the capillary-to-capsule lumina and perpendicular to the epithelia analysed?).

C) The definition of a localization (as plotted in the histograms). How did the authors cope with the large agglomerations of stain as shown in Figure 1 (and most others as well)?

3) Explain how the threshold level was set for the histogram tally of localizations relative to that used for the figures.

4) A very nice aspect of this study is the use of Tokuyasu protocol for tissue labeling. However, it would be nice to confirm some of the results using different fixation methods, as the measurements obtained are unlikely to be absolute (they are, however, excellent in terms of relative spatial positioning). Therefore, we would encourage the authors to validate some of their measurements using different fixation methods, and in live cells if at all possible.

5) In the Introduction, the authors describe the glomerular filtration barrier without reference to the podocyte slit diaphragm. As a number of kidney diseases result from the failure of this structure, the slit diaphragm should be included.

6) The assertion in the Introduction that there is little data regarding how ECM components in the basement membrane bind receptors in vivo disregards a raft of literature. We are sure that this was not the authors’ intention but more careful wording would be wise here.

7) Explain why the localization counts are so much higher for periN than NC1.

8) In the interpretation of the data on Laminin in mouse, the authors claim to show that ‘GBM's laminin component is divided into two networks, one produced by podocytes and the other by endothelial cells’. However their labelling of endogenous Laminin indicates just one central Laminin-rich region and their claim for two networks depends on data derived from the ectopic expression of human Laminin α5 from both endothelial cells and podocytes. This interpretation needs to be refined.

9) The distribution of AgrinC and integrin β1 are described as being ‘localized along the endothelial and podocyte surfaces of the human GBM’. The most striking result in Figure 5, but not mentioned by the authors, is that AgrinC is much richer on the podocyte face and almost absent from the capillary side. This discrepancy between the results presented and the text needs to be addressed.

---

## [Author Response]

*1) Explain more clearly why 3DSIM and/or STED are not be suitable for imaging 200 nm sections. The authors should also consider validating some of their key results using another imaging method, like 3DSIM and STED (and GFP fusion proteins) as it is likely to generate similar data*.

Relative molecule positioning has been accomplished using SIM, STED and STORM/PALM (references cited in our Introduction). With careful aligning and averaging particles, the precision in estimating molecule positions can be smaller than the actual resolution of the method. Thus, in principle, three dimensional structured illumination (3DSIM) and/or Stimulated Emissions Depletion (STED) microscopy would also be suitable for imaging 200 nm sections, and would have validated some of our key results. There are differences in resolution, however. The resolution of 3DSIM is around 100 nm, and the width of the GBM is around 150,200 nm, which could have obscured some of the positioning. Since we do not have access to a STED microscope, we did not compare the methods in our study.

We believe that one of our main accomplishments is a method to image endogenous protein architecture in tissue sections with EM correlation using multi-antibody labeling. The ability to image a GFP fusion protein in a mammalian tissue section would require generating a transgenic animal. In cases where good antibodies do not exist, this might be a worthwhile option. The utility of studying endogenous proteins in tissue sections rather than GFP fusion proteins opens the use of STORM to a much broader range of applications.

*2) Clarify the geometry of analysis in the text. While considerable detail is given in the methods, the reader needs to understand the geometry of analysis as they read the results. Statements such as ‘selected multiple regions across glomeruli’ are uninformative and the definition of ‘localizations’ is lacking*.

*The reader should be informed of*:

*A) How different regions were chosen (that include both capillary and podocyte epithelia in approximate cross section)*.

We wanted to avoid the mesangial regions where the mesangial matrix meets the GBM so we selected regions for analysis that were away from the mesangium, towards the most peripheral aspect of the circular capillary loops. We also avoided regions where the GBM was thicker than the known thickness of the GBM, as these areas most likely represented oblique sections. We have modified the Methods section as well as Figure 1—figure supplement 1 to graphically illustrate this point.

*B) The exact geometry of analysis (were lines joining the capillary to capsule lumina and perpendicular to the epithelia analysed?)*.

We chose regions as described above using an analytic window that was 800 nm wide. This point is noted in our Methods.

*C) The definition of a localization (as plotted in the histograms). How did the authors cope with the large agglomerations of stain as shown in*
Figure 1
*(and most others as well)*?

A STORM localization is the centroid position obtained after fitting an elliptical Gaussian distribution to the point spread function of isolated individual fluorescence intensity peaks captured in an image. This definition is now added to the Methods section.

Since each antibody bound fluorophore can blink multiple times, the agglomerations could represent multiple time resolved localizations, but could also represent multiple antibodies bound to a similar spot or multiple fluorophores bound to the same antibody. This is the most likely explanation for the “agglomerations”. Since we cannot distinguish between these possibilities and did not want to introduce any bias, we considered each localization event as an independent event. Since we are focused on relative positions, this should not have any effect on our conclusions.

*3) Explain how the threshold level was set for the histogram tally of localizations relative to that used for the figures*.

We did not set any threshold for any of the histograms. Each histogram is an accumulation of STORM localizations from multiple images, aligned to the mid point between agrin localizations, as described in the Methods.

*4) A very nice aspect of this study is the use of Tokuyasu protocol for tissue labeling. However, it would be nice to confirm some of the results using different fixation methods, as the measurements obtained are unlikely to be absolute (they are, however, excellent in terms of relative spatial positioning). Therefore, we would encourage the authors to validate some of their measurements using different fixation methods, and in live cells if at all possible*.

We tried different fixation and embedding methods and the Tokuyasu protocol gave us the best results. The traditional Tokuyasu method uses glutaraldehyde fixation and was developed to enhance cryosectioning. When we tried 2% glutaraldehyde, little staining was seen which we attribute to loss of antigenicity. Using 4% PFA/PBS fixation with Tokuyasu’s embedding method (90% sucrose/10% PVP) not only allowed us to cut thin sections, but also appeared to preserve antigenicity, as compared with 4% PFA/PBS fixation using alternate embedding media where we were unable to cut sections thinner than 1 μm. As the Tokuyasu method was developed for EM, it also allowed us to perform EM after STORM imaging.

Since our imaging was of extracellular matrix, it would not be possible to confirm this in live cells, since in vitro cultured living cells do not organize basement membranes properly.

Lastly, in our study, the calculated GBM thickness was similar to that observed using other fixation and embedding methods (traditional EM, for example). So, we are confident in our measurements. However, as noted above, we focused here on relative spatial positioning as shown in Figure 4. This is scientifically more relevant than their exact molecular positions in the GBM.

*5) In the Introduction, the authors describe the glomerular filtration barrier without reference to the podocyte slit diaphragm. As a number of kidney diseases result from the failure of this structure, the slit diaphragm should be included*.

We have modified the Introduction to include a short discussion of the podocyte slit diaphragm.

*6) The assertion in the Introduction that there is little data regarding how ECM components in the basement membrane bind receptors in vivo disregards a raft of literature. We are sure that this was not the authors’ intention but more careful wording would be wise here*.

We apologize for any confusion here. We have modified the relevant sentences in the Introduction.

*7) Explain why the localization counts are so much higher for periN than NC1*.

We don’t really know. We suspect that this could be due to differences in antibody affinities. But it could also be due to a more accessible epitope recognized by the periN antibody. We tested a variety of antibody dilutions and obtained the same result, so this is not due to differences in antibody concentration.

*8) In the interpretation of the data on Laminin in mouse, the authors claim to show that ‘GBM's laminin component is divided into two networks, one produced by podocytes and the other by endothelial cells’. However their labelling of endogenous Laminin indicates just one central Laminin rich region and their claim for two networks depends on data derived from the ectopic expression of human Laminin α5 from both endothelial cells and podocytes. This interpretation needs to be refined*.

We apologize for not making this clearer. When we used Laminin short arm or N terminal antibodies, we did observe one central Laminin rich region in mouse GBM. When we used the C terminal LG domain (human transgene derived) antibody, we identified a band of localizations on each side of this central region. We interpret both sets of data to mean that there are two Laminin layers, with the N termini either overlapping or so close to each other near the middle of the GBM that they are indistinguishable by STORM, and the C termini facing away from the center, allowing us to distinguish the two layers. The contention that there are two separate networks does rely on cell specific expression of the human Laminin α5 transgene, and we are happy to clarify how this conclusion was made. This is consistent with published work showing that both endothelial cells and podocytes synthesize Laminin α5-containing trimers (St. John et al., 2001) and that these trimers remain associated with their cellular origin (2). It is also consistent with our analysis of human GBM in Figure 5. The text has been revised to better explain this.

*9) The distribution of AgrinC and integrin β1 are described as being ‘localized along the endothelial and podocyte surfaces of the human GBM’. The most striking result in*
Figure 5*, but not mentioned by the authors, is that AgrinC is much richer on the podocyte face and almost absent from the capillary side. This discrepancy between the results presented and the text needs to be addressed*.

We thank the reviewers for bringing up this point. We consistently noted that the AgrinC staining on the podocyte side of the GBM was higher in human vs mouse. We don’t know what the explanation is, but it is possible that this is due to masking of the epitope by the collagen α1α2(IV) in the human GBM’s endothelial aspect or due to higher expression in human podocytes. We are happy to discuss this in the text.